# Influence of an Edible Oil–Medium-Chain Triglyceride Blend on the Physicochemical Properties of Low-Fat Mayonnaise

**DOI:** 10.3390/molecules27154983

**Published:** 2022-08-05

**Authors:** Heng-I Hsu, Tan-Ang Lee, Ming-Fu Wang, Po-Hsien Li, Jou-Hsuan Ho

**Affiliations:** 1Department of Food Science, Tunghai University, 1727, Sec. 4, Taiwan Boulevard, Xitun District, Taichung City 40704, Taiwan; 2Department of Food and Nutrition, Providence University, 200, Sec. 7, Taiwan Boulevard, Shalu District, Taichung City 43301, Taiwan

**Keywords:** medium-chain triglycerides, mayonnaise, physicochemical properties, rheological properties

## Abstract

Mayonnaise is a semisolid oil-in-water emulsion comprised of egg yolk, oil, and vinegar. One main problem with mayonnaise is its high fat content, so efforts have been made to develop low-fat sauces with similar characteristics to real mayonnaise. The purpose of this study was to evaluate the effect of medium-chain triglycerides (MCTs) blended with edible oil (soybean and olive oil) on the rheological, physicochemical, and sensory properties of low-fat mayonnaise. The results revealed that the shear viscosity decreased with the increase in medium-chain fatty acid (MCFA) contents and decreased with an increasing shear rate. Tan δ was <1, and a semisolid fluid with shear-thinning behavior was formed. The oscillation frequency test showed that the MCFA-containing mayonnaise was viscoelastic. The particle size and oil droplet analyses revealed that the emulsion droplet size and distribution were not significantly different in the MCT group compared to the control. The sensory evaluation demonstrated that the MCFA-containing mayonnaise was acceptable. This study illustrates that MCTs are a good substitute to produce the proper physicochemical properties of mayonnaise.

## 1. Introduction

Mayonnaise is among the oldest and most widely used sauces in the world today and is mainly prepared by mixing egg yolks, vegetable oil (soybean oil, sunflower oil, olive oil, and coconut oil), vinegar (distilled white vinegar), sugar, and salt to maintain a closely packed foam of oil droplets [1]. The total fat content in mayonnaise can be as high as 70–80%, of which the fatty acids are mainly long-chain and saturated fatty acids [2]. The consumption of low-fat food products has become more popular over the past decade. Because the amount and type of fat consumed are of importance to the etiology of several chronic diseases, such as obesity, cardiovascular diseases, and cancer, it is not surprising that consumers hold fast to nutritional guidelines concerning fat consumption. The food industry has been forced to reduce the amount of fat, sugar, cholesterol, salt, and certain additives in the diet. The development of healthy mayonnaise has become a trend because of concerns about consumer susceptibility to body health. It is possible to choose fat substitutes in specific quantities that produce a texture close to that of traditional mayonnaise.

In most European countries, the olive tree is among the main agricultural crops, principally for extra virgin olive oil (EVOO) production. The International Olive Council (IOC) has classified olive oil trees based on their purity and quality [3,4]. As such, the characterization, especially the flavor and fatty acid composition, of olive oil is very vital and widely studied [5,6,7]. Olive oil is usually characterized by various carbonyl substances, such as linear saturated and unsaturated aldehydes and alcohols, esters and hydrocarbons [8]. Furthermore, this bioactive composition of olive oil can be altered by geographical origin [9], ripening degree [7,10], extraction methods [11], cultivar and harvest year [5,12]. As mentioned before, previous studies demonstrated that olive oils contained triolein (28.60–48.25%), and significant differences in triglyceride composition based on cultivar and harvested year, even in the same geographical area, microclimatic, and agronomic conditions [5,7].

Meanwhile, due to the unique fatty acid profile, soybean oil plays a crucial role in daily life. As with olive oil, the quality of soybean oil also depends on the variety of the origin of seeds, growing condition, region, and conditions of industrial processing. Triglycerides are an essential part of the daily diet, a main source of energy, and they act as carriers of fat-soluble vitamins (A, D, E, and K) [13]. By using reverse-phase high-performance liquid chromatography (RP-HPLC) analysis, previous studies discovered 12 different fragments of triglyceride present in soybean oil, namely linoleic-linolenic-linolenic (LLnLn), linoleic-linoleic-linolenic (LLLn), linoleic-linoleic-linoleic (LLL), oleic-linoleic-linolenic (OLLn), palmitic-linoleic-linolenic (PLLn), oleic-linoleic-linoleic (OLL), palmitic-linoleiclinoleic (PLL), oleic-oleic-linoleic (OOL), palmitic-oleic-linoleic + stearic-linoleic-linoleic (POL + SLL), palmitic-palmiticlinoleic (PPL), stearic-oleic-linoleic (SOL) and oleic-oleic-oleic (OOO) [14].

Medium-chain fatty acids (MCFAs) are composed of fatty acids with 6–12 carbons [15]. Their digestion and absorption in the human body do not require cholic acids or pancreatic lipolytic enzymes for metabolism, and the lymphatic system is not required for transport; instead, MCFAs enter the liver directly via the hepatic portal vein and are rapidly β-oxidized, which increases diet-induced thermogenesis [16]. Many studies have used MCFAs in the diet to improve food function in recent years. One study demonstrated that consuming 48 g of medium-chain triglycerides (MCTs) compared to corn oil at a single time leads to a greater rise in postprandial oxygen consumption compared to the basal level [17]. Furthermore, consuming 18–24 g of MCTs for 16 continuous weeks significantly lowers endpoint body weight and fat mass in overweight men and women [18]. Thus, MCFAs can favorably reduce the accumulation of body fat and increase the expenditure and decomposition of body fat, thereby improving the effect of diet control in overweight patients. Papamandjaris et al. reported that MCFA diets improve the efficiency of adipose oxidation in healthy female adults compared with long-chain fatty acid diets, and suggested that MCFAs should be included in recommended diets to control obesity [19]. Ham et al. investigated the effect of an MCFA extract on 3T3-L1 adipocytes to explore the mechanism of regulating body weight and body fat formation. As results, MCFAs affected the proliferation of adipocytes by regulating lipoprotein lipase and lipocytic differentiation proteins, thereby regulating the production of animal body fat [20]. Previous reports have suggested that MCFAs/MCTs preserve insulin sensitivity in animal models and patients with type 2 diabetes [21].

The favorable effects of MCFAs on regulating body fat have been demonstrated in human, animal, and cellular experiments; to the best of our knowledge, only a few studies have substituted MCTs for edible oil to make mayonnaise, but the variety of processed foods made from MCFAs remain limited. Identifying a suitable fat replacer when formulating low-fat products is important because reducing fat content can deteriorate quality, leading to poor texture, mouthfeel, and flavor [22]. Plant oils and animal oils (EPA and DHA) with long-chain unsaturated fatty acids are commonly used to make mayonnaise and provide favorable characteristics [23,24,25]. Therefore, in this study, MCTs were used to prepare functional low-fat mayonnaise, and the physicochemical properties of the mayonnaise were determined.

## 2. Materials and Methods

### 2.1. Materials

Eggs, edible vinegar (distilled white vinegar), salt, sugar, soybean oil (Taiwan Sugar Co., Tainan, Taiwan; year of production: 2021; 600 mL per bottle; transparent polyethylene oil bottles), and first cold-pressed olive oil (Canoliva extra virgin olive oil, Baena, Spain; year of production: 2021; 500 mL per bottle; opaque glass oil bottles) were purchased from Taisuco, and PXMart (Taichung, Taiwan). Furthermore, MCTs of coconut oil were provided by Extra Crown GFEE International Co. Ltd. (Taipei, Taiwan) which contained 0.27% hexanoic acid (caproic acid), 48.24% octanoic acid (caprylic acid), 38.92% decanoic acid (capric acid), and 11.15% dodecanoic acid (lauric acid).

### 2.2. Preparation of the Mayonnaise

An amount of 30 g of egg yolks, 19.64 g of apple cider vinegar, 5.77 g of white granulated sugar, and 2.24 g of table salt were weighed in a beaker. The mayonnaise emulsion was prepared by slowly blending the oil with the pre-mix (water phase). The mixture was mixed evenly by stirring for 30 sec in a laboratory grade high-speed dispersion homogenizer (BS-014, Boh Sheuan Enterprise Co., Ltd., Tainan, Taiwan), at the speed of 500 rpm, and 155.88 g of blended edible oil was added according to Table 1. MCTs replaced the soybean and olive oils at levels of 50% of the total oil used. Next, the mixture was homogenized at a constant speed for 5 min. The prepared mayonnaise was packaged in sealed bags and stored at 4 °C in the dark.

### 2.3. Composition Analysis

The approximate nutritional composition of mayonnaise samples, which is the moisture, fat, protein, ash, and carbohydrate content were studied in 3 replicates and corresponded to the official methods of AOAC. The moisture content was evaluated by the hot-air oven method; ash content was analyzed by the method of incinerating samples in a muffle furnace at 550–600 °C; crude protein content was studied by the Kjeldahl method or Kjeldahl digestion method; fat content was measured by the acid hydrolysis method; cholesterol content was tested by the digitonin method; while the carbohydrate was determined by subtracting the sum of moisture, protein, fat, and ash percentages from 100%. Caloric values were calculated as: total calories = (4 × g protein) + (9 × g fat) + (4 × g carbohydrate).

### 2.4. Color Analysis

The color of the mayonnaise samples was analyzed by the Color Meter ZE-2000 (Nippon Denshku Industries, Tokyo). L* (lightness measurement), a* (greenness–redness value), and b* (blueness–yellowness value) value was studied to determine the quality changes. Calibration of the instrument involved using a standard black-and-white ceramic tile before measurement. Color measurements were carried out at room temperature in triplicate.

### 2.5. Rheological Properties

#### 2.5.1. Flow Test

A dynamic rheometer was utilized for the measurements following the method reported by Liu et al. [26]. The flow index was measured using a 40 mm stainless steel parallel plate and a 2° cone plate at a temperature of 25 °C and a plate height of 1 mm. The shear rate was set to increase from 0 to 150 (1/s) over 4 min, remain at 150 (1/s) for 4 min, and decrease from 150 to 0 (1/s) over 2 min. The Herschel–Bulkley model was used to calculate yield stress (τy), viscosity (K), the fluid behavior index (n), and thixotropic properties.

#### 2.5.2. The Oscillation Test

The method reported by Liu et al. was used as a reference [26]. First, the stress sweep was determined. The strain value in the linear viscoelastic region was set to 1%, and an oscillation test was conducted in the frequency range from 0.1 to 10 Hz. The relationships between the G’ (storage modulus) and the G” (loss modulus) and frequency were determined.

### 2.6. Particle Size Analysis

The method reported by Worrasinchai et al. was used as a reference [27]. A 150 mL aliquot of 0.1% sodium dodecyl sulfate solution was added to a beaker containing 0.04 g of mayonnaise. After the solution was mixed evenly, the opacity was adjusted to 0.2%–0.5% with deionized water, and the particle size and distribution of the sample were measured using a laser particle size analyzer at a speed of 1200 rpm, a refractive index of the sample of 1.46, and a measurement time of 10 s.

### 2.7. Emulsion Stability

The method reported by Mun et al. was modified to determine emulsion stability [28]. Five g of mayonnaise was placed in a centrifuge tube and the tube was stored in an incubator at 50 °C for 48 h. Then, the sample was centrifuged at 1600 rpm for 10 min and weighed after removing the oil layer.

### 2.8. Sensory Evaluation

Sensory analysis was conducted on the mayonnaise samples after 1 day of storage at 4 °C. Sixty students (30 males, and 30 females; age range 18 to 25 years; no smoker) from the Department of Food and Nutrition, Providence University, were selected as test evaluators to assign scores after attending a 4 h training session before the evaluation. The factors used for the evaluation included appearance, aroma, taste, greasiness, and overall acceptability, which were evaluated on a 9 points hedonic scale of 1 = the least, the lowest, and 9 = the most, the highest. The samples were arranged and coded with three-digit random numbers. Furthermore, the serving order was completely randomized. The contents of the evaluation form were fully explained before the evaluation, and warm water and soda crackers were provided as palate cleansers.

### 2.9. Statistical Analysis

Experimental assessments and analyses were carried out in triplicate. Experimental data were analyzed with Duncan’s new multiple range test using SAS software (SAS Institute, Cary, NC, USA) to compare the differences between treatment. A *p*-value < 0.05 was considered significant. All experiments were performed in triplicate, and data are expressed as the mean ± SD.

## 3. Results and Discussion

### 3.1. Chemical Composition and Caloric Values

The chemical composition and the caloric values of the mayonnaise samples are listed in Table 2. Moisture content increased when MCTs were substituted compared to soybean and olive mayonnaise, which may have been due to the water solubility of the MCTs. MCTs differ from long-chain triglycerides (LCTs) as they are relatively soluble in water and, thus, are quickly hydrolyzed and absorbed [29]. According to previous studies, the dispersed phase must decrease and water content must increase to stabilize the emulsion and increase the viscosity of light mayonnaise [30,31]. However, the carbohydrate content of MCT containing mayonnaise was higher than that of mayonnaise is prepared with soybean oil and olive oil. This result corresponded to a previous study that examined the properties of low-fat mayonnaise with different fat mimetics based on whey protein isolate and pectin [32]. Moreover, the fat content of MCTs in the mayonnaise was much lower than that in soybean and olive mayonnaise. The caloric values of MCT-replacement mayonnaise were significantly lower (*p* < 0.05) than soybean and olive mayonnaise because different oil contents are added to soybean oil, and olive oil mayonnaise [32]. MCTs provide about 10% fewer calories than LCTs, or 8.4 calories/gram for MCTs vs. 9.2 calories/gram for LCTs [33]. MCTs are fats found in foods, such as soybean and coconut oil. MCTs are metabolized differently than the LCTs found in most other foods. Unlike longer-chain fatty acids, MCTs are transported directly to the liver, where they are used as an instant energy source or turned into ketones [34]. Ketones are substances produced when the liver breaks down large amounts of fat. Due to these properties, MCT has been researched for its benefits to exercise performance and health.

### 3.2. Color Analysis of Low-Fat Mayonnaise

Table 3 provides the color analysis of the low-fat mayonnaise prepared with different oils and MCTs. The physical appearance of a product affects consumer choice and the intention to purchase in several ways [35]. Hence, it is important to determine a suitable formulation for producing mayonnaise. The *L**, *a**, and *b** values were determined to analyze changes in the quality of the mayonnaise. *L** represents the lightness measurement; *a** characterizes the greenness–redness value, while *b** considers the blueness–yellowness value. The lightness of the mayonnaise decreased slightly after substituting MCTs into the mayonnaise. The *L** value of the soybean oil mayonnaise decreased from 87.24 to 86.86 after MCT was added. The *L** value of the olive oil mayonnaise decreased from 87.29 to 86.72 (olive + MCT). Replacing MCT in the oil phase of the mayonnaise tended to increase the yellowness (*b** value) of the final products. The *b** value of MCT was 23.57. The *b** value increased from 21.98 (soybean oil mayonnaise) to 25.94 (soybean + MCT mayonnaise), and 21.24 (olive oil mayonnaise) to 25.01 (olive + MCT mayonnaise), respectively, after substituting MCTs. Substituting MCTs decreased the *a** value of the soybean oil mayonnaise from 2.03 to 1.82. In contrast, the *a** value increased from 0.77 to 0.88 in the olive oil mayonnaise. 

### 3.3. Rheological Properties of Functional Low-Fat Mayonnaise

Figure 1 demonstrates the viscosity changes in the mayonnaise samples prepared with different oils/MCTs at different shear rates (s^−1^). The results revealed that the viscosities of the five groups of samples decreased with increasing shear rate and formed a shear-thinning fluid, confirming that the mayonnaise was between a plastic fluid and a pseudoplastic fluid. The viscosity in each group was examined; the olive oil group (olive) had the highest viscosity, followed by the soybean oil group (soybean) and the MCT group. Further observations of the functional low-fat mayonnaises revealed that the viscosity of the half soybean oil + half MCT group (soy + MCT) was slightly lower than that of the soybean group. The same result was found for the half olive oil + half MCT group (olive + MCT) and the olive group. We speculate that these results are related to the low viscosity of MCTs. Although adding MCTs caused a slight decrease in the viscosity of the sample, the viscosity of the olive + MCT group was close to that of the soybean group, and the viscosity of the soy + MCT group was higher than that of the MCT group. Interestingly, the apparent viscosity of the mixed oil samples emulsions increased sequentially as the amount of MCTs added was increased. The 20% sample had the largest apparent viscosity, which was due to the increase in saturated fatty acid content in the oil phase of the emulsion as the proportion of MCTs was increased. The emulsion oil droplets partially crystallized, which increased the viscosity [36].

Figure 2 shows the changes in the G’ and G” values during oscillation from a frequency of 0.1 to 10 Hz. The results show that the G’ and G” values for the five groups had an upward trend with increasing oscillation frequency. Further observations of the G’ and G” values for the five groups showed that the G’ values were greater than the G” values, indicating that the five groups had viscoelasticity. When the energy stored by a sample is greater than the energy lost, the state of the material is closer to a solid. Similar semisolid results were reported by Liu et al. for wheat gluten [32], by Sun et al. for whey protein–pectin complex [37], and by Chang et al. for fat substitutes [38], such as protein particles and pectin in low-fat mayonnaise. 

A substance is closer to a solid state when the tan δ value is less than 1. The tan δ values for the five groups were all less than 1 (Figure 3), further supporting that the functional low-fat mayonnaise was semisolid. Similar semisolid results were described by Liu et al. for wheat gluten [26], by Sun et al. for whey protein–pectin complex [37], and by Chang et al. for fat substitutes [38], such as protein particles and pectin in low-fat mayonnaise.

The olive group had the highest τy value in the fluid parameter analysis (Table 4), followed by the olive + MCT group and the soybean group. The MCT group had the lowest τy value. Further examination of the τy values after replacing the raw oils with MCTs revealed that the τy value for the soybean group was larger than that for the soy + MCT group and the τy value for the olive group was larger than that for the olive + MCT group. Therefore, adding MCTs decreased the τy value of the samples. The force required for a sample to flow is relatively small as the τy value decreases; therefore, it is closer to a liquid state. However, opposite results were observed for the K value. All the examined mayonnaise samples revealed thixotropic shear-thinning behavior, as their flow properties depended on the shear rate and time. The carbon chain length of MCTs is shorter than ordinary fatty acids, possibly resulting in relatively dense emulsion droplets during emulsification.

### 3.4. Particle Size Measurements of the Functional Low-Fat Mayonnaise

The particle size distribution analysis of the low-fat mayonnaise (Figure 4) revealed that the emulsion droplets of the soybean and MCT groups exhibited consistent distributions. In contrast, the soybean + MCT group, the olive oil group, and the olive oil + MCT group exhibited coalesced, inhomogeneous distributions of droplets, and an emulsion droplet distribution >10 μm. Similar phenomena have been reported previously using MCTs in rice bran oil mayonnaise [15]. 

The emulsion droplet distribution data and average volume were further examined (Table 5), and the values for the soybean oil group and the MCT group at d (0.1), d (0.5), and d (0.9) were significantly lower than in the soybean + MCT group. This finding indicates that the soybean and MCT groups possessed small particle sizes and that the mean volume D [4,3] of the emulsion droplets was significantly smaller (*p* < 0.05). MCTs and long-carbon-chain unsaturated fatty acids were used in the soybean + MCT group. The difference in fatty acid chain length led to uneven emulsification of the mayonnaise in the samples. Further research is needed to study this phenomenon. Worrasinchai et al. used soybean oil and the unimodal particle size distribution was the same as that observed in this experiment. The average particle size in the mayonnaise was 9.09 μm, which agreed with that measured in this study (8.37 μm) and corresponded to the value reported by Liu et al., who used pectin as a fat substitute in mayonnaise [27,32].

### 3.5. Emulsion Stability of the Functional Low-Fat Mayonnaise

Figure 5 shows the low-fat mayonnaise with different oils at the optimum ratios. The emulsion stability of the mayonnaise samples in the soybean oil group containing MCTs (84.47%) decreased significantly after adding MCTs (*p* < 0.05) compared with the soybean oil mayonnaise (91.51%). A previous study concluded that 65 and 78% of soybean oil mayonnaise has a shelf life of 40 and 32 days compared to groundnut oil-based mayonnaise because the lecithin content of soybean oil improves the emulsion stability of the mayonnaise [39].

The emulsion in the olive + MCT group was slightly unstable (79.86%) compared with the high emulsion stability of the olive group. Di Mattia et al. and Giacintucci et al. showed that adding olive oil causes uneven emulsions in mayonnaise and that the main cause is phenolic compounds in the olive oil, particularly oleuropein [40,41]. After 14 days of storage, the mayonnaise was affected by the external conditions and its emulsion structure, and the emulsion structure destabilized, resulting in a significant decrease in the emulsion stability of all samples. The emulsion stabilities of the soybean and olive groups were maintained well, while the groups containing MCTs had lower emulsion stabilities (79.42% for 0 days; 66.09% for 14 days); their emulsion stabilities significantly decreased by 15.38% (soy + MCT), 7.94% (olive + MCT), and 13.33% (MCTs). These results show that the emulsion stability of the mayonnaise decreased slightly after using MCTs as a replacement.

### 3.6. Sensory Evaluation

Table 6 reveals the sensory evaluation results (appearance, aroma, taste, greasiness, and overall acceptability) for the mayonnaise products with different oils and MCTs. The highest appearance score (5.82 and 6.00 points) was found in the soybean and olive samples, but no significant differences were observed among the MCT groups (*p* > 0.05). The appearance of the olive oil group was closer to that of the soybean group, while the MCT group was closer to liquid and was more like a sauce, leading to higher acceptability compared to the MCT groups. No significant difference was observed in aroma (*p* > 0.05). The MCT groups received the highest taste scores of 4.95, and the other groups were not significantly different, but the average scores were higher than that of the soybean and olive groups (3.97 and 3.03 points). This may have occurred because apple cider vinegar was used to make the low-fat mayonnaise, giving it a light apple flavor, and making it more palatable to consumers. The soybean and olive groups had a significantly lower oily/greasy feeling score than that of the other groups because of the high oil content. The other groups were not significantly different (*p* > 0.05). The MCT group had an overall acceptability score of 5.03, while the olive oil group received relatively low scores of 4.08. We speculate that olive oil has a grassy flavor making it less acceptable.

## 4. Conclusions

The results of this study showed that mayonnaise can be prepared by substituting MCTs. The sample slightly shifted toward a fluid state after adding the MCTs, but the difference was not obvious. The particle size and microstructural analyses showed that the emulsion droplet size and distribution were not significantly different in the soybean + MCT group compared to in the soybean oil group. Adding MCTs generally did not affect the overall acceptability of the mayonnaise in the sensory evaluation. It was concluded that mayonnaise can be successfully produced from soybean oil by substituting MCTs to produce low-fat mayonnaise. Further studies on the stability and conditions of homogenization, and simultaneous deaeration of the mayonnaise should be carried out. Meanwhile, storage stability of the MCT-substituted mayonnaise may be studied to determine the shelf life of products.

## Figures and Tables

**Figure 1 molecules-27-04983-f001:**
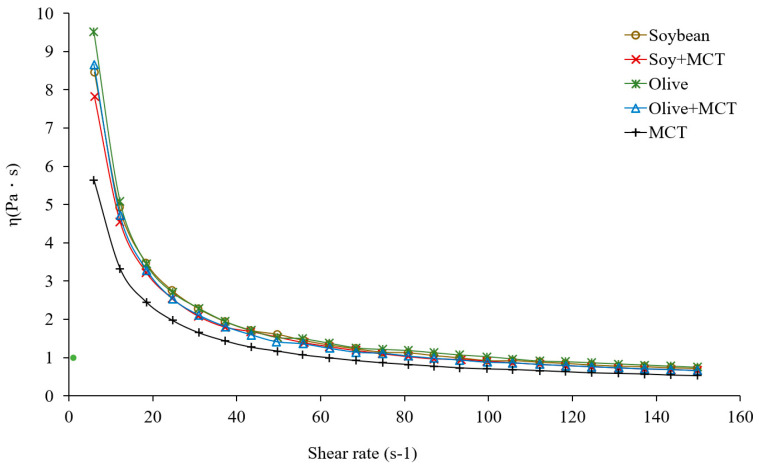
Steady shear viscosity curve of the mayonnaise samples. (○) Soybean oil; (×) soybean oil + medium-chain triglycerides; (✽) olive oil; (△) olive oil + medium-chain triglycerides; and (+) medium-chain triglycerides.

**Figure 2 molecules-27-04983-f002:**
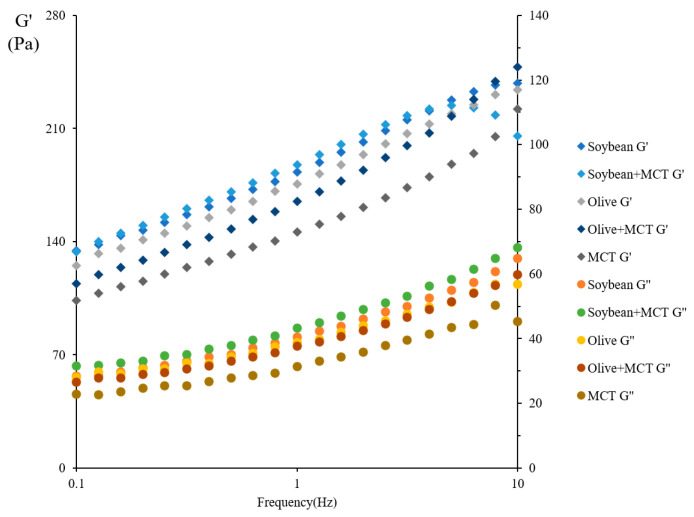
Effect of frequency on the G’ and G” values of the mayonnaise samples.

**Figure 3 molecules-27-04983-f003:**
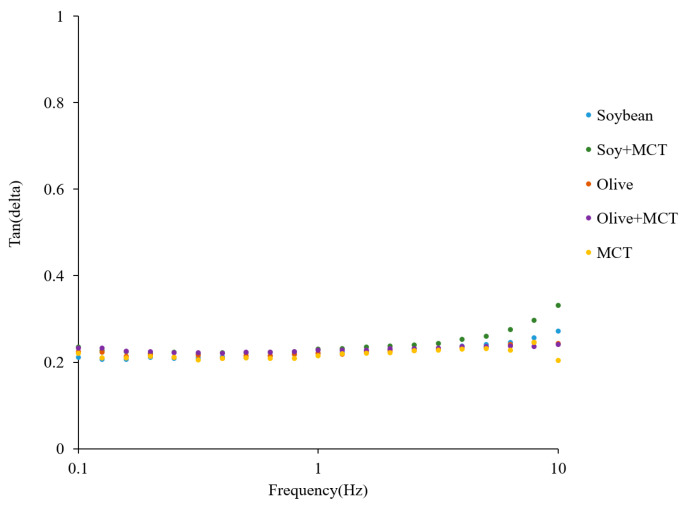
Effect of frequency on the tan(delta) of the mayonnaise samples.

**Figure 4 molecules-27-04983-f004:**
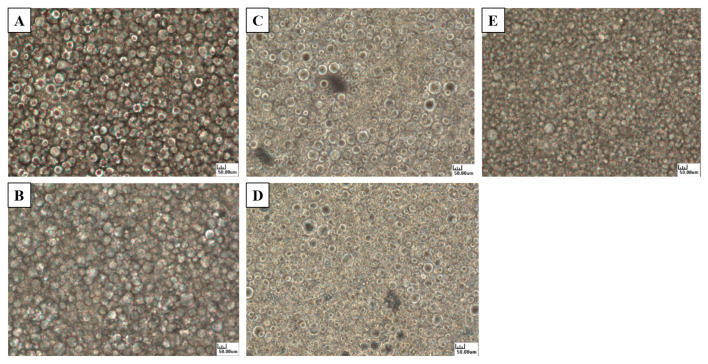
Optical micrographs of the mayonnaise samples. (**A**) Soybean oil; (**B**) soybean oil + MCTs; (**C**) olive oil; (**D**) olive oil + MCTs; (**E**) MCTs.

**Figure 5 molecules-27-04983-f005:**
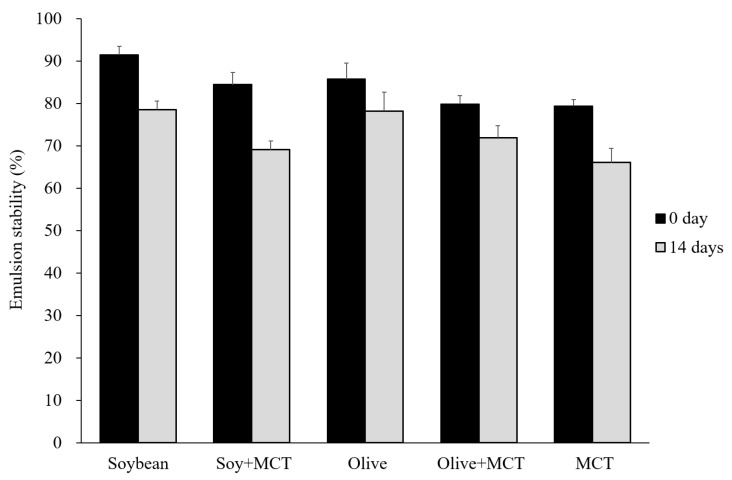
Emulsion stability of the mayonnaise samples at the optimum ratios. Mean ± SD values followed by the same letter in each bar are not significantly different (*p* > 0.05) (n = 3).

**Table 1 molecules-27-04983-t001:** Mayonnaise formulation.

Samples	Soybean	Soybean + MCT	Olive	Olive + MCT	MCT
Egg yolk (%)	14.0	14.0	14.0	14.0	14.0
Vinegar (%)	9.3	9.3	9.3	9.3	9.3
Sugar (%)	2.7	2.7	2.7	2.7	2.7
Salt (%)	1.0	1.0	1.0	1.0	1.0
Soybean oil (%)	73.0	36.5	-	-	-
MCT (%)	-	36.6	-	36.5	73.0
Olive oil (%)	-	-	73.0	36.5	-

**Table 2 molecules-27-04983-t002:** Chemical composition analysis (%, *w/w*) and caloric values of the mayonnaise samples.

Sample	Moisture	Fat	Protein	Ash	Carbohydrate	Caloric Value *
Soybean	37.13 ± 0.05 ^c^	77.94 ± 0.17 ^a^	2.21 ± 0.07	1.17 ± 0.06	2.12 ± 0.19 ^d^	715.90 ^a^
Soybean + MCT	52.48 ± 0.23 ^a^	40.95 ± 0.13 ^b^	2.14 ± 0.13	1.23 ± 0.02	3.20 ± 0.13 ^b^	389.94 ^b^
Olive	38.53 ± 0.29 ^c^	76.29 ± 0.31 ^a^	2.19 ± 0.06	1.19 ± 0.04	2.26 ± 0.25 ^d^	704.41 ^a^
Olive + MCT	50.85 ± 0.37 ^b^	40.22 ± 0.27 ^b^	2.13 ± 0.11	1.18 ± 0.02	3.09 ± 0.24 ^c^	381.29 ^b^
MCT	52.14 ± 0.05 ^a^	41.32 ± 0.18 ^b^	2.15 ± 0.06	1.25 ± 0.05	3.34 ± 0.15 ^a^	393.05 ^b^

^a–d^ Means with different letters within the same column differed significantly (*p* < 0.05). * Caloric value = (fat × 9) + (protein × 4) + (carbohydrates × 4).

**Table 3 molecules-27-04983-t003:** Color analysis of the mayonnaise samples.

Sample	*L**	*a**	*b**	ΔE
Soybean	87.24 ± 0.00 ^a^	2.03 ± 0.01 ^a^	21.98 ± 0.02 ^d^	-
Soybean + MCT	86.86 ± 0.01 ^b^	1.82 ± 0.03 ^b^	25.94 ± 0.07 ^a^	3.91
Olive	87.29 ± 0.01 ^a^	0.77 ± 0.01 ^d^	21.24 ± 0.04 ^e^	1.46
Olive + MCT	86.72 ± 0.01 ^b^	0.88 ± 0.03 ^c^	25.01 ± 0.05 ^b^	3.28
MCT	87.10 ± 0.01 ^a^	−2.35 ± 0.02 ^e^	23.57 ± 0.06 ^c^	4.66

^a–e^ Means with different letters within the same column differed significantly (*p* < 0.05).

**Table 4 molecules-27-04983-t004:** Model-fitting flow equation parameters and thixotropy of the mayonnaise samples.

Sample	τ_y_ (Pa)	K (Pa s^n^)	n	Thixotropy (Pa/s)
Soybean	37.81 ± 4.99 ^b^	4.65 ± 0.78 ^c^	0.52 ± 0.02 ^b^	2249 ± 382 ^e^
Soybean + MCT	29.49 ± 3.54 ^c^	9.90 ± 1.90 ^b^	0.40 ± 0.03 ^c^	2766 ± 237 ^a^
Olive	45.47 ± 1.07 ^a^	2.38 ± 0.28 ^e^	0.65 ± 0.01 ^a^	2289 ± 176 ^d^
Olive + MCT	45.14 ± 1.27 ^a^	3.46 ± 0.57 ^d^	0.57 ± 0.03 ^b^	2588 ± 201 ^b^
MCT	14.55 ± 2.32 ^d^	10.97 ± 0.92 ^a^	0.36± 0.01 ^c^	2460 ± 106 ^c^

^a–d^ Means with different letters within the same column differed significantly (*p* < 0.05).

**Table 5 molecules-27-04983-t005:** Distribution of the oil droplets and volume mean diameter in the mayonnaise samples.

Samples	d (0.1) (μm)	d (0.5) (μm)	d (0.9) (μm)	D [4,3] (μm)
Soybean	3.43 ± 0.16 ^a^	6.01 ± 0.05 ^a^	10.41 ± 0.38 ^c^	14.73 ± 2.78 ^a^
Soybean + MCT	2.60 ± 0.03 ^b^	5.64 ± 0.02 ^b^	12.07 ± 0.60 ^a^	8.73 ± 2.69 ^b^
Olive	3.54 ±0.07 ^a^	6.24 ± 0.12 ^a^	11.25 ± 0.12 ^b^	14.52 ± 2.13 ^a^
Olive + MCT	2.51 ± 0.03 ^c^	5.53 ± 0.04 ^b^	11.07 ± 0.23 ^b^	8.75 ± 1.79 ^b^
MCT	2.78 ± 0.02 ^b^	5.41 ± 0.10 ^bc^	10.10 ± 0.40 ^c^	8.96 ± 3.09 ^b^

^a–c^ Means with different letters within the same column differed significantly (*p* < 0.05).

**Table 6 molecules-27-04983-t006:** Sensory evaluation of the mayonnaise samples.

Sample	Appearance	Aroma	Taste	Greasiness	Overall Acceptability
Soybean	5.82 ± 1.23 ^a^	4.50 ± 1.41 ^a^	3.97 ± 1.57 ^c^	4.02 ± 1.42 ^b^	4.33 ± 1.40 ^c^
Soybean + MCT	4.80 ± 1.06 ^b^	4.35 ± 1.35 ^ab^	4.40 ± 1.56 ^b^	4.87 ± 1.26 ^a^	4.73 ± 1.41 ^ab^
Olive	6.00 ± 1.29 ^a^	4.57 ± 1.33 ^a^	3.03 ± 1.71 ^d^	4.07 ± 1.54 ^b^	4.08 ± 1.52 ^d^
Olive + MCT	4.90 ± 1.14 ^b^	4.55 ± 1.22 ^a^	4.37 ± 1.56 ^b^	4.78 ± 1.30 ^a^	4.57 ± 1.44 ^b^
MCT	5.10 ± 0.99 ^b^	4.67 ± 1.18 ^a^	4.95 ± 1.50 ^a^	4.95 ± 1.40 ^a^	5.03 ± 1.44 ^a^

^a–d^ Means with different letters within the same column differed significantly (*p* < 0.05).

## Data Availability

The datasets used and/or analyzed during the current study are available from the corresponding author on request.

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
