# Peer review of "Influence of an Edible Oil–Medium-Chain Triglyceride Blend on the Physicochemical Properties of Low-Fat Mayonnaise"

_molecules, 2022, doi:10.3390/molecules27154983_

Round 1

Reviewer 1 Report

The reviewed article concerned the evaluation of the effect of medium-chain triglycerides (MCTs) mixed with edible oil (soybean and olive oil) on the rheological, physicochemical and sensory properties of low-fat mayonnaise. The results obtained in the study are interesting and, after conducting further research, they can be used in industry. Manuscript is well written, but I have some comments. If they are taken into account by the authors, the article may be published. See details below.

Introduction

Line 42: according to many literature data, medium chain fatty acids are also fatty acids with 6 to 12 carbon atoms in the molecule, please check and correct it.

Line 57: what does the abbreviation MVFAs stand for? Is it not a mistake?

Materials and Methods

Materials: please provide the composition of the fat blend containing medium-chain fatty acids. What fatty acids were included in its composition, what oil was used?

Methods

Preparation of the mayonnaise: The method of preparation of mayonnaise differs from the methods used in industry for low- and high-fat mayonnaise. A prerequisite for high stability of mayonnaise emulsions is adequate homogenization to obtain small size of the fat globules, combined with the simultaneous deaeration of the emulsion during homogenization. The description in the paper shows that no homogenization and deaeration were used during the preparation of the emulsion.

The composition of mayonnaise given in Table 1 should be given in % [g of each ingredient i 100 g of mayonnaise]. Please change it.

Particle size analysis: Was an appropriate method used to measure the particle size of the fat present in the emulsion? Due to the delicate consistency of fat globules, which may merge into larger clusters or disintegrate under the influence of heat or the addition of water, the microscopic method of measuring their size seems to be more advantageous.

There is no description of the methods used to measure the color and chemical composition of the designed mayonnaises.

Results and Discussion

Chemical composition and caloric values

Table 2: it is not necessary to mark the results that do not differ statistically significantly with the same letters (results regarding ash and protein content)

Color analysis of low-fat mayonnaise

Table 3: Please check letters a-f next to results that differ significantly. There are incorrectly assigned letters.

Rheological properties of functional low-fat mayonnaise

What do the G' and G'' values mean?

Table 4, 5: Please check letters a-f next to results that differ significantly. There are incorrectly assigned letters.

Sensory evaluation

Table 6: Please check letters a-f next to results that differ significantly. There are incorrectly assigned letters.

Conclusion: please add in this chapter that further studies on the stability and conditions of homogenization and with simultaneous deaeration should be carried out and tests on the stability of mayonnaise in a long, e.g. 6-month storage period should be carried out.

Author Response

We feel great thanks for your professional review work on our article. As you are concerned, there are several problems that need to be addressed. According to your nice suggestions, we have made extensive corrections to our previous draft, the detailed, and specific changes we have made in the revised manuscript are, as follows:

Reviewer 1

Answers or explanations

Introduction

Line 42: according to many literature data, medium chain fatty acids are also fatty acids with 6 to 12 carbon atoms in the molecule, please check and correct it.

Thanks for the positive comment. After checked for the reference, the “8 to 10” carbon atoms had been re-corrected into “6 to 12”. The changes had been marked in yellow background in the manuscript.

Line 57: what does the abbreviation MVFAs stand for? Is it not a mistake?

Thank you for the comment. The word “MVFAs” had been re-corrected into “MCFAs”. All changes are marked in yellow background in the revised manuscript.

Materials and Methods

Materials: please provide the composition of the fat blend containing medium-chain fatty acids. What fatty acids were included in its composition; what oil was used?

Thanks for the positive comment. The MCTs of coconut oil was provided by Extra Crown GFEE International Co. Ltd (Taipei, Taiwan) which contained 0.27% hexanoic acid (caproic acid), 48.24% octanoic acid (caprylic acid), 38.92% decanoic acid (capric acid), and 11.15% dodecanoic acid (lauric acid). The corrected sentence was marked in yellow background in the revised manuscript.

Preparation of the mayonnaise: The method of preparation of mayonnaise differs from the methods used in industry for low- and high-fat mayonnaise. A prerequisite for high stability of mayonnaise emulsions is adequate homogenization to obtain small size of the fat globules, combined with the simultaneous deaeration of the emulsion during homogenization. The description in the paper shows that no homogenization and deaeration were used during the preparation of the emulsion.

Thank you for the comment. The mayonnaise sample was homogenized by a laboratory grade high-speed dispersion homogenizer (BS-014, Boh Sheuan Enterprise co., Ltd., Tainan, Taiwan). The dry ingredients were mixed and dissolved in vinegar by mixing the solution for 5 min. The oil was added in gradually while mixing continued up to 10 min.

References:
1. Akhtar, G.; Masoodi, F.A. Structuring functional mayonnaise incorporated with Himalayan walnut oil Pickering emulsions by ultrasound assisted emulsification. Ultrasonics Sonochemistry 2022, 86, 106022.

2. Hakimian, F.; Emamifar, A.; Karami, M. Evaluation of microbial and physicochemical properties of mayonnaise containing zinc oxide nanoparticles. LWT 2022, 163, 113517.

The composition of mayonnaise given in Table 1 should be given in % [g of each ingredient in 100 g of mayonnaise]. Please change it.

Thank you for the positive evaluation. The g of each ingredient in 100 g of mayonnaise was changed to %.

Particle size analysis: Was an appropriate method used to measure the particle size of the fat present in the emulsion? Due to the delicate consistency of fat globules, which may merge into larger clusters or disintegrate under the influence of heat or the addition of water, the microscopic method of measuring their size seems to be more advantageous.

There is no description of the methods used to measure the colour and chemical composition of the designed mayonnaises.

Thank you for the positive comment. Particle size analysis was performed by using a laser diffractometer Mastersizer 2000 with the Hydrosizer 2000S module (Malvern Instruments, UK). The sample was extemporaneously dispersed in purified water at 2500 rpm until an obscuration rate was obtained. Dilution may induce physicochemical changes in the environment of the droplets, leading to completely unreliable data. Therefore, measurements on a commercial standardized emulsion were carried out.

Reference:

1. Worrasinchai, S.; Suphantharika, M.; Pinjai, S.; Jamnong, P. β-Glucan prepared from spent brewer's yeast as a fat replacer in mayonnaise. Food Hydrocolloids 2006, 20, 68-78.

The methods used to analyse the colour and chemical composition of the sample was added in section 2.3 and 2.4. The corrected sentence was marked in yellow background in the revised manuscript.

Results and Discussion

Chemical composition and caloric values

Table 2: it is not necessary to mark the results that do not differ statistically significantly with the same letters (results regarding ash and protein content)

Due to there is statistically significant different in the results regarding the moisture, fat, carbohydrates, and caloric value which tabulated in the same table, hence, the ash and protein content in the same tables also mark with the letters.

Color analysis of low-fat mayonnaise

Table 3: Please check letters a-f next to results that differ significantly. There are incorrectly assigned letters.

Thank you for your comment. The letters next to the results in Table 3 was re-corrected. The changes had been marked in yellow background in the manuscript.

Rheological properties of functional low-fat mayonnaise

What do the G' and G'' values mean?

Table 4, 5: Please check letters a-f next to results that differ significantly. There are incorrectly assigned letters.

Thanks for the comment. The rheological properties were presented in Figure 2. The rheological properties, while the storage modulus (G') and loss modulus (G')' as a function of oscillatory stress. The letters next to the results in Table 4 and 5 was re-corrected. The changes had been marked in yellow background in the manuscript.

Sensory evaluation

Table 6: Please check letters a-f next to results that differ significantly. There are incorrectly assigned letters.

Thank you for your positive reviewed. The letters next to the results in Table 6 was re-corrected. The changes had been marked in yellow background in the manuscript.

Conclusion

Please add in this chapter that further studies on the stability and conditions of homogenization and with simultaneous deaeration should be carried out and tests on the stability of mayonnaise in a long, e.g. 6-month storage period should be carried out.

Thanks for the assessment. The conclusion section had been added in the further studies that may be carry out for the MCT substituted mayonnaise.

Reviewer 2 Report

Manuscript Number: molecules-1828485, titled: Influence of an edible oil- medium-chain triglyceride blend on the physicochemical properties of low-fat mayonnaise

Review 1 – 12 July 2022

I suggest a major revision

To the Authors (in detail):

1.  This manuscript needs to be improved mainly in the introduction section. The authors have to better describe the state of the art and the soy-bean and olive oil compositions. The M&M section has to be improved and detailed. The bibliography has to be improved and extended in the introduction section. Inaccuracies in the text. The manuscript is not organised as per instructions for Authors of Molecules

2.  The manuscript is not organized as per instructions for authors of Molecules, verify the sequence of sections and sub-sections;

3.     Introduction section, before ref.1: which type of vinegar?

4.     Introduction section, before ref1: what type of vegetable oils?

5.    Introduction section, detail short-chain and long-chain triglycerides. You will write about long-chain triglycerides later;

6. Introduction section, generally, medium-chain fatty acids are considered 6-12 fatty acids, please, verify a more wide bibliography;

7.     Introduction section, extend the discussion comparing different types of vegetable oils and fatty acid composition;

8.     Introduction section, page 2: what is MVFA? Please, verify carefully; write the complete words before the abbreviation;

9. Introduction section, extend the state of the art with essential fatty acid (EFA) content in edible vegetable oils, mainly in relation with soybean oil olive oil (i.e. the oils of your experiment);

10. Introduction section, your study is related with soybean oil and olive oil but no information is given about their triglyceride composition of soybean oil [X1 – X2] and olive oil [X3-X4]. Indicate the olive oil composition and the factors influencing the composition such as: geographical area of production, cultivar, harvest date, harvest year. Find, read and discuss the following references. Do not cumulate references at the end of the sentence but include references after each type of oil:

[X1] Processing impact on tocopherols and triglycerides composition of soybean oil and its deodorizer distillate evaluated by high-performance liquid chromatography.

Turk J Chem. 2020; 44(6): 1694–1702. doi: 10.3906/kim-2005-10

[X2] Determination and Comparison of Seed Oil Triacylglycerol Composition of Various Soybeans (Glycine max (L.)) Using H-1-NMR Spectroscopy.

Molecules 18(11):14448-14454 (2013). DOI: 10.3390/molecules181114448

[X3] Variation in triacylglycerols of olive oils produced in Calabria (Southern Italy) during olive ripening

Riv. Ital. Sostanze Gr. 91 (4), 221-240 (2014).

[X4] Influence of cultivar and harvest year on triglyceride composition of olive oils produced in Calabria (Southern Italy).

European Journal of Lipid Science and Technology, 115 (8) 928-934 (2013)

DOI: 10.1002/ejlt.201200390.

11. 2.1 sub-section, no information is given about the soybean oil and the olive oil. There is a lot of information to include in relation with year of production, type of container (material and color); quantity of oil per bottle; oil geographical origin; which type of olive oil? Indicate all information existing in the label;

12. Your manuscript focus on olive oil but you have not explained what olive oil is. Please, discuss about the different categories of olive oil listed by the European Regulation for olive oil, pages 11-12-13-14, Annex I. If you read “olive oil” on the label, the bottle contains the category 5 of the above mentioned table [X5]. The same is in the by the International Olive Oil Council Trade standard [X6]. Explain if this is olive oil for you or not. Please, be clear.

[X5] COMMISSION REGULATION (EEC) No 2568/91 of 11 July 1991 on the characteristics of olive oil and olive-residue oil and on the relevant methods of analysis, amended by 01991R2568 — EN — 04.12.2016 — 031.005 — 1,

 [X6] TRADE STANDARD APPLYING TO OLIVE OILS  AND OLIVE POMACE OILS. COI/T.15/NC No 3/Rev. 12, June 2018

13. 2.1 section: what type of vinegar?

14. 2.1 subsection, detail how MCTs were prepared,  their origin and their physical state;

15. 2.2 sub-section: the whole egg? Detail;

16. Table 1: did you use water?

17. 2.3.1 sub-section, Liu et al 2018 and in the whole manuscript. Please verify how to include the bibliography. Use the template and compare with some recently published papers;

18. 2.6 sub-section: detail how many male and how many females. In addition: how many smokers? The range of years of the panelists; students of Food school or what?

19. 2.7 and 3.1 sub-section, caption of table 2 and in the whole manuscript: when you indicate the statistical significance, verify the spacing between letter, symbol and numeric value, verify also if capital or small letter and if italicized or not;

20. 3.5 sub-section, soybean oil (be consistent with your text);

21. 3.5 sub-section, replace Di et al with Di Mattia et al;

22. The references section is not arranges as per the instructions for Authors of Molecules (MDPI), for example the journal name has to be abbreviated with the proper abbreviation, see> Google > Journal abbreviation list;

23. Ref 19 and in the whole manuscript, when you write a scientific name you have to apply the International binomial regulation: write in italics;

24. Please, write in blue color or evidence differently the corrections you will do

I suggest a major revision

Regards.

Author Response

We feel great thanks for your professional review work on our article. As you are concerned, there are several problems that need to be addressed. According to your nice suggestions, we have made extensive corrections to our previous draft, the detailed, and specific changes we have made in the revised manuscript are, as follows:

Reviewer 2

Answers or explanations

Introduction

1. This manuscript needs to be improved mainly in the introduction section. The authors have to better describe the state of the art and the soybean and olive oil compositions. The M&M section has to be improved and detailed. The bibliography has to be improved and extended in the introduction section. Inaccuracies in the text. The manuscript is not organised as per instructions for Authors of Molecules

Thank you for the comments. All the changes in the revised manuscript had been written by using blue colour words and marked in yellow background.

2. The manuscript is not organized as per instructions for authors of Molecules, verify the sequence of sections and sub-sections.

Thanks for the assessment. The sequence of sections and sub-sections of the manuscript had been rearranged.

3. Introduction section, before ref.1: which type of vinegar?

Thank you for the comment. The sentence had been rewritten and wrote in blue colour in the manuscript.

4. Introduction section, before ref1: what type of vegetable oils?

Thank you for the reviewed. The sentence had been rewritten and wrote in blue colour in the manuscript.

5. Introduction section, detail short-chain and long-chain triglycerides. You will write about long-chain triglycerides later;

Thank you very much for the positive evaluation and references suggestions. The “Introduction” section has been re-written and redraft with the appropriate content based on the reviewer’s comment. All changes are written by using the blue colour words in the revised manuscript.

6. Introduction section, generally, medium-chain fatty acids are considered 6-12 fatty acids, please, verify a wider bibliography;

Thanks for the positive comment. After checked for the reference, the “8 to 10” carbon atoms had been recorrected into “6 to 12”. The changes had been marked in yellow background in the manuscript.

7. Introduction section, extend the discussion comparing different types of vegetable oils and fatty acid composition;

Thank you very much for the positive evaluation and references suggestions. The “Introduction” section has been re-written and redraft with the appropriate content based on the reviewer’s comment. All changes are written by using the blue colour words in the revised manuscript.

8. Introduction section, page 2: what is MVFA? Please, verify carefully; write the complete words before the abbreviation;

Thank you for the comment. The word “MVFAs” had been recorrected into “MCFAs”. All changes are marked in yellow background in the revised manuscript.

9. Introduction section, extend the state of the art with essential fatty acid (EFA) content in edible vegetable oils, mainly in relation with soybean oil olive oil (i.e. the oils of your experiment);

Thank you very much for the positive evaluation and references suggestions. The “Introduction” section has been re-written and redraft with the appropriate content based on the reviewer’s comment. All changes are written by using the blue colour words in the revised manuscript.

10. Introduction section, your study is related with soybean oil and olive oil but no information is given about their triglyceride composition of soybean oil [X1 – X2] and olive oil [X3-X4]. Indicate the olive oil composition and the factors influencing the composition such as: geographical area of production, cultivar, harvest date, harvest year. Find, read and discuss the following references. Do not cumulate references at the end of the sentence but include references after each type of oil:

[X1] Processing impact on tocopherols and triglycerides composition of soybean oil and its deodorizer distillate evaluated by high-performance liquid chromatography.

Turk J Chem. 2020; 44(6): 1694–1702. doi: 10.3906/kim-2005-10

[X2] Determination and Comparison of Seed Oil Triacylglycerol Composition of Various Soybeans (Glycine max (L.)) Using H-1-NMR Spectroscopy.

Molecules 18(11):14448-14454 (2013). DOI: 10.3390/molecules181114448

[X3] Variation in triacylglycerols of olive oils produced in Calabria (Southern Italy) during olive ripening

Riv. Ital. Sostanze Gr. 91 (4), 221-240 (2014).

[X4] Influence of cultivar and harvest year on triglyceride composition of olive oils produced in Calabria (Southern Italy).

European Journal of Lipid Science and Technology, 115 (8) 928-934 (2013)

DOI: 10.1002/ejlt.201200390.

Thank you very much for the positive evaluation and references suggestions. The “Introduction” section has been re-written and redraft with the appropriate content based on the reviewer’s comment. All changes are written by using the blue colour words in the revised manuscript.

Materials and Methods

11. 2.1 sub-section, no information is given about the soybean oil and the olive oil. There is a lot of information to include in relation with year of production, type of container (material and color); quantity of oil per bottle; oil geographical origin; which type of olive oil? Indicate all information existing in the label.

Thanks for the positive evaluation. The sub-section 2.1. had been rewritten and added in the relevant information by using the blue colour words in the manuscript.

12. Your manuscript focus on olive oil but you have not explained what olive oil is. Please, discuss about the different categories of olive oil listed by the European Regulation for olive oil, pages 11-12-13-14, Annex I. If you read “olive oil” on the label, the bottle contains the category 5 of the above-mentioned table [X5]. The same is in the by the International Olive Oil Council Trade standard [X6]. Explain if this is olive oil for you or not. Please, be clear.

[X5] COMMISSION REGULATION (EEC) No 2568/91 of 11 July 1991 on the characteristics of olive oil and olive-residue oil and on the relevant methods of analysis, amended by 01991R2568 — EN — 04.12.2016 — 031.005 — 1,

[X6] TRADE STANDARD APPLYING TO OLIVE OILS  AND OLIVE POMACE OILS. COI/T.15/NC No 3/Rev. 12, June 2018

Thanks for the reviewer’s comment. The olive oil used in this study was first cold press olive oil (Canoliva extra virgin olive oil, Baena, Spain; year of production: 2021; 500 mL per bottle; opaque glass oil bottles) were purchased from PXMart (Tai-chung, Taiwan).

13. 2.1 section: what type of vinegar?

Thanks for the positive evaluation. The sub-section 2.1. had been rewritten and added in the relevant information by using the blue colour words in the manuscript.

14. 2.1 subsection, detail how MCTs were prepared, their origin and their physical state.

Thanks for the positive comment. The MCTs of coconut oil was provided by Extra Crown GFEE International Co. Ltd (Taipei, Taiwan) which contained 0.27% hexanoic acid (caproic acid), 48.24% octanoic acid (caprylic acid), 38.92% decano-ic acid (capric acid), and 11.15% dodecanoic acid (lauric acid). The corrected sentence was marked in yellow background in the revised manuscript.

15. 2.2 sub-section: the whole egg? Detail.

Only egg yolk had been used to prepare the mayonnaise sample in this study.

16. Table 1: did you use water?

No water has use in this study to prepare the mayonnaise.

17. 2.3.1 sub-section, Liu et al 2018 and in the whole manuscript. Please verify how to include the bibliography. Use the template and compare with some recently published papers.

Thank you for the comment. The reference style and template had been referred to arrange the format of the manuscript.

18. 2.6 sub-section: detail how many male and how many females. In addition: how many smokers? The range of years of the panelists; students of Food school or what?

Thank you for the positive evaluation. Section 2.8 (previous 2.6 sub-section) had been re-corrected and re-written in blue colour in the manuscript.

Results and Discussion

19. 2.7 and 3.1 sub-section, caption of table 2 and in the whole manuscript: when you indicate the statistical significance, verify the spacing between letter, symbol and numeric value, verify also if capital or small letter and if italicized or not.

Thank you for your comment. The letters next to the results in all tables was re-corrected. The changes had been marked in yellow background in the manuscript.

20. 3.5 sub-section, soybean oil (be consistent with your text);

Thank you for the comment. The word “soya” had been changed to “soybean”.

21. 3.5 sub-section, replace Di et al with Di Mattia et al.

Thank you. The reference Di et al had been changed to Di Mattia et al.

22. The references section is not arranges as per the instructions for Authors of Molecules (MDPI), for example the journal name has to be abbreviated with the proper abbreviation, see> Google > Journal abbreviation list.

Thanks for the reviewed. Every reference cited in the text was also presented in the reference list. The references in manuscript had been tabulated and arranged by using the reference management software (EndNote).

23. Ref 19 and in the whole manuscript, when you write a scientific name you have to apply the International binomial regulation: write in italics.

Thank you for the assessment. All the scientific name in the manuscript and references had been changed to italics.

24. Please, write in blue color or evidence differently the corrections you will do

Thank you. All the changes in the revised manuscript had been written by using blue colour words and marked in yellow background.

Round 2

Reviewer 1 Report

Dear Authors, thank you for considering my comments on the manuscript. I have one more remark about the revised version of the work, namely table 2: the results regarding the content of protein and ash do not show statistically significant differences (all values are with the index a, in such a situation there is no need to add the letter "a" to the obtained results).

Please also check for stylistic errors through the whole manuscript, including changing the capital letter P, when indicating statistically significant differences, to a lower case p.

Author Response

We feel great thanks for your professional review work on our article. As you are concerned, there are several problems that need to be addressed. According to your nice suggestions, we have made extensive corrections to our previous draft, the detailed, and specific changes we have made in the revised manuscript are, as follows:

Reviewer 1

Answers or explanations

Table 2: the results regarding the content of protein and ash do not show statistically significant differences (all values are with the index a, in such a situation there is no need to add the letter "a" to the obtained results).

Thank you for your comment. The letters next to the results in Table 2 was deleted. The changes had been marked in yellow background in the manuscript.

Please also check for stylistic errors through the whole manuscript, including changing the capital letter P, when indicating statistically significant differences, to a lower case p.

Thank you for the positive evaluation. The capital letter P, when indicating statistically significant differences, was changed to a lower-case p.

Reviewer 2 Report

Manuscript Number: molecules-1828485, titled:

Influence of an edible oil- medium-chain triglyceride blend on the physicochemical properties of low-fat mayonnaise

Review 2 – 24 July 2022

To the Authors (in detail):

  1. The authors have included many of my comments, anyway some correction more is necessary. The references section has to be carefully revised.
  2. The manuscript is not always organized as per instructions for authors of Molecules, 2.7., 3.3, sub-sections and in the whole manuscript, when you indicate the bibliography, you have to apply the instructions for authors of Molecules. Do not include the publication year after the name of the author;
  3. When you have indicated the statistical significance you have used different criteria: p<0.05 (p in small letter, not italicized and no spaces between letter, symbol and numeric value, caption of table 5); P < 0.05 (P in capital letter and italicized with spaces between letter, symbol and numeric value, 3.6 sub-section);
  4. References section, ref 8, the names of authors are wrong, you have changed the first names with the family names. Write: Esposto, S.; Montedoro, G.; Selvaggini, R.; Riccò, I.; Taticchi, A.; Urbani, S.; Servili, M.
  5. References section: refs 6-7, in the first name of the author, change: A with A.M.;
  6. References section: the journals names are almost all incorrect. You have to apply the instructions for authors of Molecules, please, read them carefully and use some recently published paper as a template;
  7. References section, the journal name has to be abbreviated. Go to Google and digit: abbreviation and the journal name. After this, write the abbreviation in the manuscript in italics;
  8. References section: ref 6, the correct abbreviation is not LWT, please, verify;
  9. References section, ref 9 and in the whole section, after the page range, do not write a comma but a dot;
  10. References section, ref 11, the abbreviation of the journal is wrong and it has to be written in small letters;
  11. Ref 14: Uddin in capital letters? Verify the guidelines;
  12. Ref 25 and in the whole section, please, do not underline the doi number;
  13. Refs 32-33 and in the whole section, sometime you have written the title in capital letter and sometime in small letters. Please, be consistent in the whole section;
  14. Ref 25, there is an asterisk after… choice. Why?
  15. Ref 41: Chem in capital letter.
  16. Please, write in red color or evidence differently the corrections you will do

Regards.

Author Response

We feel great thanks for your professional review work on our article. As you are concerned, there are several problems that need to be addressed. According to your nice suggestions, we have made extensive corrections to our previous draft, the detailed, and specific changes we have made in the revised manuscript are, as follows:

Reviewer 2

Answers or explanations

The manuscript is not always organized as per instructions for authors of Molecules, 2.7., 3.3, sub-sections and in the whole manuscript, when you indicate the bibliography, you have to apply the instructions for authors of Molecules. Do not include the publication year after the name of the author.

Thank you for the comment. All the publication year after the name of authors was deleted.

When you have indicated the statistical significance you have used different criteria: p<0.05 (p in small letter, not italicized and no spaces between letter, symbol and numeric value, caption of table 5); P < 0.05 (P in capital letter and italicized with spaces between letter, symbol and numeric value, 3.6 sub-section).

Thank you for the positive evaluation. The capital letter P, when indicating statistically significant differences, was changed to a lower-case p.

References section, ref 8, the names of authors are wrong, you have changed the first names with the family names. Write: Esposto, S.; Montedoro, G.; Selvaggini, R.; Riccò, I.; Taticchi, A.; Urbani, S.; Servili, M.

References section: refs 6-7, in the first name of the author, change: A with A.M.;

Thanks for the comment. Reference 6,7, and 8 had been re-corrected.

References section: the journals names are almost all incorrect. You have to apply the instructions for authors of Molecules, please, read them carefully and use some recently published paper as a template;

Thanks for the reviewed. Every reference cited was re-corrected and re-written in the manuscript.

References section, the journal name has to be abbreviated. Go to Google and digit: abbreviation and the journal name. After this, write the abbreviation in the manuscript in italics;

Thank you for the assessment. All the reference cited in this manuscript was re-corrected and re-written.

References section: ref 6, the correct abbreviation is not LWT, please, verify;

Thanks for the comment. Reference 6 had been re-corrected.

References section, ref 9 and in the whole section, after the page range, do not write a comma but a dot;

Thank you for the reviewed. Reference 9 and the following references used in this manuscript had been re-corrected and re-written.

References section, ref 11, the abbreviation of the journal is wrong and it has to be written in small letters;

Thank you. The abbreviation of reference 11 and the other references used in this manuscript had been re-corrected and re-written.

Ref 14: Uddin in capital letters? Verify the guidelines;

Thank you for the positive evaluation. The author’s name had been changed from “UDDIN” to “Sirajuddin”.

Ref 25 and in the whole section, please, do not underline the doi number;

Thank you for the comment. The doi number of reference 25 and the other references used in this manuscript had been re-corrected and re-written.

Refs 32-33 and in the whole section, sometime you have written the title in capital letter and sometime in small letters. Please, be consistent in the whole section;

Thanks for the reviewed. Every reference cited was re-corrected and re-written in the manuscript.

Ref 25, there is an asterisk after… choice. Why?

Thank you for the reviewed. Every reference cited was re-corrected and re-written in the manuscript.

Ref 41: Chem in capital letter.

Thank you for the assessment. All the references cited in the manuscript had been re-corrected and re-written.